# Analysis of Volatile Components in *Rosa roxburghii* Tratt. and *Rosa sterilis* Using Headspace–Solid-Phase Microextraction–Gas Chromatography–Mass Spectrometry

**DOI:** 10.3390/molecules28237879

**Published:** 2023-11-30

**Authors:** Yuhang Deng, Huan Kan, Yonghe Li, Yun Liu, Xu Qiu

**Affiliations:** 1Key Laboratory of Forest Resources Conservation and Utilization in the Southwest Mountains of China Ministry of Education, Southwest Forestry University, Kunming 650224, China; dengyuhang0503@swfu.edu.cn (Y.D.); kanhuan@swfu.edu.cn (H.K.); 2Forest Resources Exploitation and Utilization Engineering Research Center for Grand Health of Yunnan Provincial Universities, Kunming 650224, China; 3College of Plant Protection, Yunnan Agricultural University, Kunming 650201, China; swfclyh@126.com

**Keywords:** *Rosa roxburghii* Tratt., *Rosa sterilis*, flavor characteristics, odor threshold, HS-SPME-GC-MS

## Abstract

Volatile organic compounds (VOCs) and flavor characteristics of *Rosa roxburghii* Tratt. (RR) and *Rosa sterilis* (RS) were analyzed using headspace solid-phase microextraction coupled with gas chromatography–mass spectrometry (HS-SPME-GC-MS). The flavor network was constructed by combining relative odor activity values (ROAVs), and the signature differential flavor components were screened using orthogonal partial least squares discriminant analysis (OPLS-DA) and random forest (RF). The results showed that 61 VOCs were detected in both RR and RS: 48 in RR, and 26 in RS. There were six key flavor components (ROAVs ≥ 1) in RR, namely nonanal, ethyl butanoate, ethyl hexanoate, (*3Z*)-3-hexen-1-yl acetate, ethyl caprylate, and styrene, among which ethyl butanoate had the highest contribution, whereas there were eight key flavor components (ROAVs ≥ 1) in RS, namely 2-nonanol, (*E*)-2-hexenal, nonanal, methyl salicylate, β-ocimene, caryophyllene, α-ionone, and styrene, among which nonanal contributed the most to RS. The flavor of RR is primarily fruity, sweet, green banana, and waxy, while the flavor of RS is primarily sweet and floral. In addition, OPLS-DA and RF suggested that (*E*)-2-hexenal, ethyl caprylate, β-ocimene, and ethyl butanoate could be the signature differential flavor components for distinguishing between RR and RS. In this study, the differences in VOCs between RR and RS were analyzed to provide a basis for further development and utilization.

## 1. Introduction

*Rosa roxburghii* Tratt. (RR) and *Rosa sterilis* (RS) are deciduous shrubs of the genus *Rosa* in the family Rosaceae. RR is rich in vitamin C (Vc), superoxide dismutase (SOD), organic acids, minerals, and polysaccharides and is known as the “King of Vc” [1,2,3,4]. Modern pharmacological studies have found that RR has a variety of physiological activities, such as delaying aging [5], improving immunity [6], lowering blood sugar and blood lipids [7], and pre-detoxification [8]. RS was discovered in Guizhou, China, in 1985 [9], whose fruit is golden yellow, and the surface of which is basically free of thorns. RS displays physiological activities similar to those of RR [9]. However, RS has a thicker flesh, moderate acidity, and higher flavonoid and polyphenol contents than RR [10].

Volatile organic compounds (VOCs) can affect the flavor of fruits, attract animals to spread seeds [11], and also have antimicrobial properties that help to prolong the storage time of fruits [12]. Flavor is an essential characteristic of VOCs in fruit, the intensity of which influences the acceptance and purchasing desire of the consumer [13]. RR and RS, as third-generation fruits, are usually mixed and processed for sale as juices, jams, wines, and other products due to their sour taste and similar properties. However, the proportion of RR and RS paired in the products lacks criteria; thus, the analysis of signature difference flavor components between RR and RS is crucial and can contribute to the quality control and development of new products. Recently, studies on flavor components have mainly focused on RR from Guizhou province, China [13,14], while fewer studies have been conducted on the differences in flavors between RR and RS [10].

Headspace solid-phase microextraction combined with gas chromatography–mass spectrometry (HS-SPME-GC-MS) is a green and environmentally friendly analytical technique with the advantages of high sensitivity, rapidity, simplicity of operation, and high reproducibility, which is widely used to detect VOCs [15,16,17,18]. Azam et al. have analyzed VOCs from flowers in different citrus flowering stages and the leaves of different citrus types using HS-SPME-GC-MS and found that fully open citrus flowers had the highest number of VOCs [19], and that VOCs in leaves of different citrus types were correlated with developmental stage and genetic type [20]. Hu et al. have analyzed the effects of Saccharomyces cerevisiae and non-Saccharomyces cerevisiae on citrus wines using HS-SPME-GC-MS and found that mixed fermentation could improve the flavor quality of citrus wines [21]. In addition, HS-SPME-GC-MS has also been used to characterize the VOCs in Yunnan *Luculia* at different developmental stages [22] and the VOCs in wild roses at different flowering stages [23].

RR and RS have great potential as green plant resources of medicinal and edible origins. Recently, studies on RR and RS have mainly focused on food processing, active ingredients, and their functions [24,25,26,27,28]. In this study, the VOCs in RR and RS were determined using HS-SPME-GC-MS, and their signature differential flavor components were screened using principal component analysis (PCA), orthogonal partial least squares discrimination analysis (OPLS-DA), and random forest (RF). The objective of this study was to elucidate the differences in VOCs between RR and RS and to identify the signature difference in flavor components, providing data support for the exploitation and quality control of RR and RS.

## 2. Results

### 2.1. VOCs in RR and RS

The relevant information and relative contents of the VOCs in RR and RS are shown in Table 1. As shown in Table 1 and Figure 1, 61 VOCs were detected in RR and RS, including 48 in RR and 26 in RS, with 13 common components. The structures of the detected VOCs were classified into nine categories: alcohols (4), ethers (1), aldehydes (3), acids (3), esters (10), alkanes (1), terpenoids (28), aromatics (9), and others (2). The highest relative content of terpenoids was found in RR (43.10%), followed by esters (30.83%), while aldehydes were predominant in RS, followed by terpenoids with relative contents of 51.40% and 21.42%, respectively. In addition, RR was more enriched in VOCs than in RS.

The relative contents of VOCs were clustered using a heat map, as shown in Figure 2A, and the 61 VOCs were classified into four categories. Group I consisted of seven species, including *Z,Z,Z*-1,5,9,9-tetramethyl-1,4,7-cycloundecatriene, valencene, which had a high content in RR and little or none in RS; Group II consisted of 49 species, such as nonanal and benzaldehyde, which were low in both RR and RS; Group III contained (*E*)-2-hexenal and hexanoic acid, with a high content in RS and little or none in RR; and Group IV contained selina-4,11-dien, caryophyllene, and styrene, with a high content in both RR and RS. As shown in Table 1, (*E*)-2-hexenal accounted for 47.88% of the VOCs in RS, suggesting that (*E*)-2-hexenal may be the key flavor component of RS. In addition, δ-cadinene and ethyl acetate accounted for 16.16% and 14.46% of the VOCs in RR, respectively, indicating that δ-cadinene and ethyl acetate may be the key flavor components of RR.

As shown in Figure 2B, PCA showed that the variance contribution of PC1 and PC2 to VOCs reached 72.0%, indicating that the two main components could represent the main flavor characteristics of RR and RS and that the two samples were well differentiated.

### 2.2. ROAVs Analyses in RR and RS

VOCs can only be perceived when a threshold is reached, thus affecting the fruit flavor. The ROAV is a calculation that relies on a threshold of VOCs, and the ROAVs size is proportional to the intensity of the aroma, which is widely utilized for the calculation of various fruit flavors [29]. To further distinguish the VOCs in RR and RS, the dataset was narrowed using ROAVs, and the key flavor components with ROAVs ≥ 1 were selected for analysis. Subsequently, ROAVs with the same odor descriptions were summed to construct a flavor network. As shown in Table 2 and Figure 3A,B, the flavor of RR was mainly enriched in fruity, sweet, green banana, and waxy, and the key flavor components were ethyl butanoate, ethyl hexanoate, nonanal, ethyl caprylate, (*3Z*)-3-hexen-1-yl acetate, and styrene, with ethyl butanoate contributing the most to the flavor of RR. The flavor of RS was mainly sweet and floral, and the key flavor components were nonanal, styrene, (E)-2-hexenal, caryophyllene, α-ionone, β-ocimene, 2-nonanol, and methyl salicylate, with nonanal contributing the most.

### 2.3. Screening of Signature Difference Flavor Components

RF is a commonly used feature selection method that ranks the importance of key flavor components based on the Gini coefficient, where the larger the Gini coefficient, the higher the importance [31]. The relative contents of the key flavor components in RR and RS were substituted into an online website (https://cloud.oebiotech.cn) to obtain their Gini coefficients. As shown in Figure 4, the relatively more important key flavor components in RR and RS were (*E*)-2-hexenal, ethyl caprylate, ethyl butanoate, methyl salicylate, and β-ocimene, according to the Gini coefficient.

OPLS-DA can exclude irrelevant data by orthogonalization, facilitating the screening of signature differential flavor components between RR and RS [32]. As shown in Figure 5A, RR and RS were clearly distinguished in the OPLS-DA score plot with R^2^X-R^2^Y < 0.3 and Q^2^ > 0.5, indicating that the model fitted the parameters well and possessed a strong predictive ability. In addition, the cross-validation results revealed that the intercepts of the Q^2^ and Y-axis were less than zero (Figure 5B), suggesting that the OPLS-DA model did not overfit and could be used for data analyses. Therefore, in the present study, the variable importance in the projection (VIP) values of the key flavor components in RR and RS was calculated based on the OPLS-DA model.

VIP ≥ 1 and *X_A_* > 0.5 for VOCs can be used as criteria for determining them as signature difference flavor components [33]. The VIP and *X_A_* values of the key flavor components of RR and RS are shown in Table 3. The results indicated that the signature difference flavor components between RR and RS were (*E*)-2-hexenal, ethyl caprylate, β-ocimene, and ethyl butanoate, which fulfilled the conditions of VIP ≥ 1 and *X_A_* > 0.5.

## 3. Discussion

VOCs are the primary source of flavor, whose types and proportions play a decisive role in fruit flavor [34]. Humans perceive odors through G-protein-coupled odorant receptors in the olfactory epithelial cells of the nasal cavity interacting with VOCs. However, VOCs can only be perceived and recognized by the human body when a certain threshold is reached, thus affecting the human body’s judgment of fruit flavor [35]. In this study, RR had much higher VOCs than RS and was dominated by terpenoids followed by esters, whereas RS was dominated by aldehydes followed by terpenoids. The flavor of RR is mainly fruity, sweet, green banana, and waxy, while the flavor of RS is primarily sweet and floral. Zhao et al. found that the VOCs content of RR from Anshun, Guizhou Province, China, was higher than that of RS. However, the main VOCs in both RR and RS were esters [36], unlike in the present study, which may have been due to differences in sample sources and analytical methods.

Aldehydes and esters mainly originate from the oxidative breakdown of fatty acids or amino acids, presenting relatively low thresholds and significantly impacting fruit flavor, and are major contributors to fruit flavor [16]. The effect of aldehydes on fruits is dominated by the composition of the overall combined aldehyde, which negatively affects the flavor of fruit juices if there is a high level of lipid-derived aldehydes and conversely increases the fruity flavor of the fruits [37]. It was found that fermentation with lactic acid bacteria could reduce most of the lipid-derived aldehydes [37], implying that lactic acid bacteria fermentation can be used to reduce the negative impact of aldehydes on flavor in the production of RR- and RS-related products. Notably, benzaldehyde and (*E*)-2-hexenal were the primary aldehydes detected in RS, while (*E*)-2-hexenal was also the signature difference flavor components between RR and RS. Benzaldehyde, which may be produced from phenylalanine by the combined action of aminotransferase, oxygen, and manganese, is a key aldehyde affecting the flavor of fruits with its pleasant flavor [37]. In addition, as a natural green leaf volatile with pungent vegetable and green fruit flavors, (*E*)-2-hexenal contributes to the overall flavor of fruits and reduces pests and diseases [38]. In vivo and in vitro assays have also shown that (*E*)-2-hexenal can be used as a potentially efficient and eco-friendly antifungal fumigant to protect peanut seeds from the contamination of A. flavus during storage [39,40]. Herein, the high (*E*)-2-hexenal content in RS suggests that RS may be more resistant to pests and diseases than RR.

As an important flavor component, esters usually provide fruity flavors. Studies have shown that ester biosynthesis requires two substrates, acyl-CoA molecules and alcohols produced by the catabolism of amino acids or fatty acids, and is affected by various enzymes and amino acids in metabolic pathways [34,41]. In the present study, esters, the second most important category in RR, were less abundant in RS, suggesting that the fruity flavors of RR are more prominent than RS. Ethyl butyrate and ethyl caprylate, the signature difference flavor components between RR and RS, were detected only in RR. Ethyl butyrate has a flavor similar to kiwi and pineapple [42], while ethyl caprylate has a fruit flavor similar to banana [43]. And they are commonly used in flavor production.

Terpenoids are critical secondary metabolites with low flavor thresholds and characteristic flavors that can help attract pollinators and seed dispersers [44]. As typical terpenoids, triterpenoids and sesquiterpenes have physiological activities such as anticancer, antiviral, and antibacterial [45]. Among them, sesquiterpenes are also functional precursors for synthesizing fragrances, biofuels, and pharmaceuticals and are produced by sesquiterpene synthases in the cytosol [46]. δ-cadinene is the most abundant terpenoid in RR. Studies have shown that δ-cadinene has significant acaricidal activity against *Psoroptes cuniculi* in vitro [47]. In addition, β-ocimene was the signature difference flavor components between RR and RS and was detected only in RR detected only in RS. The research found that β-ocimene was significantly increased in infested fruits and may have biocontrol effects [48]. Moreover, previous research suggests that β-ocimene also possesses promising in vitro antileishmania activity [49].

## 4. Materials and Methods

### 4.1. RR and RS Samples

The fresh RR and RS were both harvested on 22 October 2022 from Aziying Town (102°45′18″ N, 25°3′51″ E), Kunming, Yunnan Province, China, and then preserved in a −80 °C refrigerator until analyses.

The top soils of RR and RS were rinsed with sterile water, dried in the shade, and pulped using a pulper (HR 2037, Philips Home Appliances Investment Co., Shanghai, China), and 5 mL of the homogenate was placed in a 20 mL headspace vial.

### 4.2. HS-SPME Conditions

The solid-phase fiber extraction head (50/30 μm DVB/CAR/PDMS, Supelco, Bellefonte, PA, USA) was aged in the GC inlet at 250 °C for 30 min. The headspace vial was fixed on the SPME device and heated at 50 °C for 10 min, and then the aged extraction head was inserted and adsorbed at 50 °C for 20 min for GC injection detection. Each sample was analyzed four times.

### 4.3. GC-MS Conditions

GC (7890B, Agilent Technologies, Santa Clara, CA, USA) conditions: HP-5MS column (30 m × 0.25 mm × 0.25 μm), carrier gas of He, flow rate of 1.0 mL/min^−1^, inlet temperature of 250 °C. The ramp-up procedure was as follows: initial temperature was set at 60 °C, held for 2 min, and then the temperature increased to 180 °C at a rate of 4 °C/min and was held for 3 min. Injection method: no-split injection.

MS (7000D, Agilent Technologies, Santa Clara, CA, USA) conditions: electronic impact (EI) of 70 eV, interface temperature of 280 °C, ion source temperature of 230 °C, mass range of 30–500 *m*/*z*, solvent delay time of 5.0 min, and full scan mode.

### 4.4. Qualitative Analyses of GC-MS

The NIST.14 L mass spectrometry database was used for the analysis and identification of VOCs, and results with a match >80 were selected to calculate the relative content of each component using the area normalization method.

### 4.5. Calculation of Relative Odor Activity Value

The relative odor activity value (*ROAV*) can be used to evaluate the contribution of individual *VOCs* to the overall flavor. The *ROAV* ranges between 0 and 100, where *VOCs* with *ROAVs* ≥ 1 are considered the key flavor-contributing compounds and *VOCs* with 0 < *ROAVs* < 1 are considered the flavor modifiers [29,50]. The *ROAV* is calculated as follows:(1)$$C=\frac{VOCs\;peak\;area}{Total\;VOCs\;peak\;area}$$
(2)$$OAV=\frac{C}{T}$$
(3)$$ROAV=\frac{OAV_{i}}{OAV_{max}}\times 100$$
where *C* is the relative content of *VOCs* (%), *T* is the odor threshold of the compound in water (mg/kg) and is taken from a book titled “*Compilations of odor threshold values in air, water and other media*”, *OAV* is the odor activity value of the compound, *OAV_max_* is the highest odor activity value, and *OAV_i_* is the lowest odor activity value.

### 4.6. Calculation of OPLS-DA and RF

OPLS-DA was established using the software SIMCA-P 14.1 to rank the key flavor-contributing components based on VIP [51]; RF, which was performed with the assistance of an online website (https://cloud.oebiotech.cn (12 September 2023)), and the Gini index (Gini) were used to rank the key flavor-contributing components [31]. A linear function normalization method was applied to normalize the VIP and Gini values, and their mean values (*X_A_*) were calculated. Moreover, VIP ≥ 1 and *X_A_* > 0.5 were employed as screening criteria for signature differential flavor components [33]. The formula is as follows:(4)$$X_{Vnom}=\frac{X_{V}-V_{min}}{V_{max}-V_{min}}$$
(5)$$X_{Gnom}=\frac{X_{G}-G_{min}}{G_{max}-G_{min}}$$
(6)XA=XVom+XGom2
where *X* is the specific key flavor contributing compound, *X_Vnom_* is the normalized value of VIP, *X_V_* is the VIP value of *X*, *V_max_* and *V_min_* are the maximum and minimum values in the VIP ranking, *X_Gnom_* is the normalized value of Gini, *X_G_* is the Gini value of *X*, *G_max_* and *G_min_* are the maximum and minimum values in the Gini ranking, and *X_A_* is the average of *X_Vnom_* and *X_Gnom_*.

### 4.7. Statistical Analyses

Excel 2019 (Microsoft, New York, NY, USA) was used to perform statistical analyses and calculations on experimental data. Origin 2021 (Origin Lab, Northampton, MA, USA) was used to plot histograms, Venn plots, and heat maps. Simca 14.1 (Umetrics, Umea, Sweden) was utilized for the PCA, OPLS-DA, and plotting.

## 5. Conclusions

In the present study, HS-SPME-GC-MS was used to detect RR and RS VOCs, and a total of 61 VOCs species were detected, of which 48 were found in RR and 26 in RS, with a total of 13 common components. Terpenoids were dominant in RR followed by esters, while aldehydes were dominant in RS followed by esters. According to ROAVs, six key flavor components (ROAVs ≥ 1) were detected in RR, namely ethyl butanoate, ethyl hexanoate, nonanal, ethyl caprylate, (*3Z*)-3-hexen-1-yl acetate, and styrene, with ethyl butanoate contributing the most to the flavor in RR, whereas eight key flavor components (ROAVs ≥ 1) were identified in RS, namely nonanal, styrene, (*E*)-2-hexenal, caryophyllene, α-ionone, β-ocimene, 2-nonanol, and methyl salicylate, among which nonanal provided the greatest flavor contribution. The flavor of RR is mainly fruity, sweet, green banana, and waxy, while the flavor of RS is mainly sweet and floral. Additionally, analyses of the key flavor components using OPLS-DA and RF revealed that (*E*)-2-hexenal, ethyl caprylate, β-ocimene, and ethyl butanoate can be used as the signature difference flavor components to distinguish RS from RR. The present investigation identified and screened the signature difference flavor components between RR and RS to provide data support for the development and quality control of RR and RS. However, VOCs may vary with conditions during processing, which may affect product quality; therefore, further research into the effects of different processing methods on the flavor compositions of RR and RS is necessary.

## Figures and Tables

**Figure 1 molecules-28-07879-f001:**
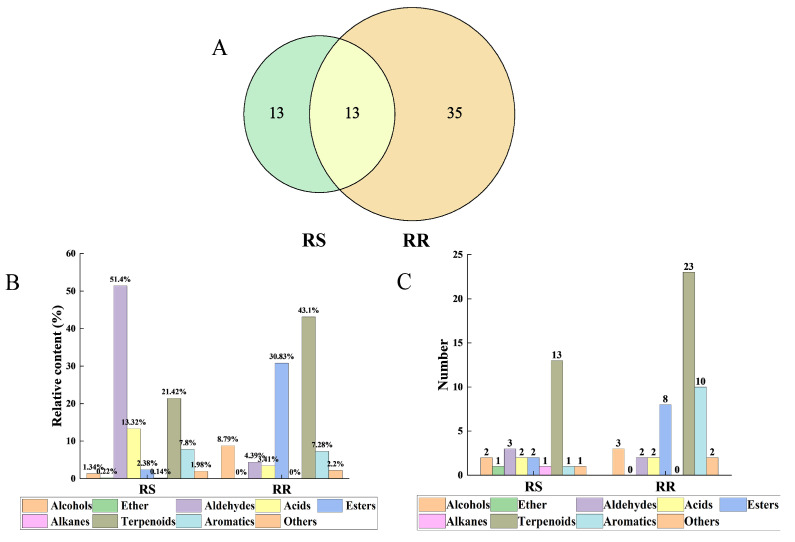
Comparison of VOCs between RR and RS. (**A**) Venn diagram of VOCs; (**B**) relative content of VOCs; (**C**) number of VOCs.

**Figure 2 molecules-28-07879-f002:**
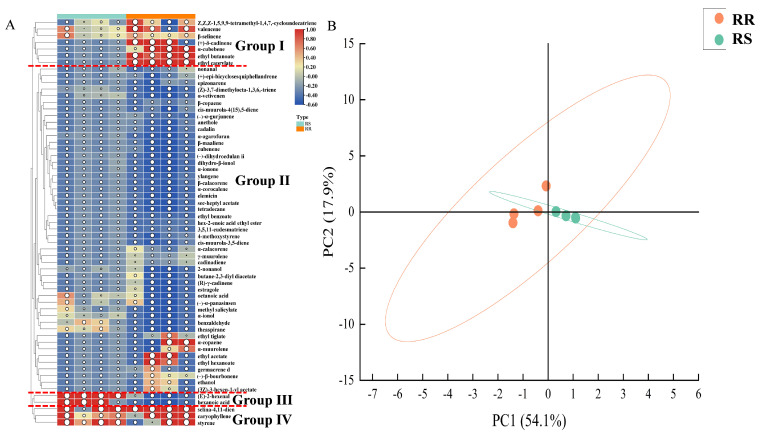
Comparison of the differences in VOCs between RR and RS. (**A**) Heat map of the VOCs; the relative content of VOCs is indicated by the color and the size of the circle, where blue indicates low content, red indicates high content, and the size of the circle indicates intensity. (**B**) PCA of VOCs.

**Figure 3 molecules-28-07879-f003:**
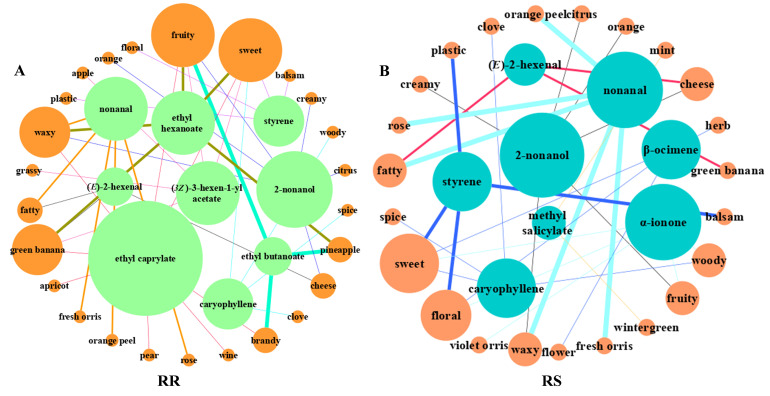
ROAVs flavor network. (**A**) ROAVs flavor network of RR; (**B**) ROAVs flavor network of RS. External nodes represent odor description and internal nodes represent key flavor components; the size of the circle indicates the number of connected edges, and the thickness of the line indicates the ROAVs size.

**Figure 4 molecules-28-07879-f004:**
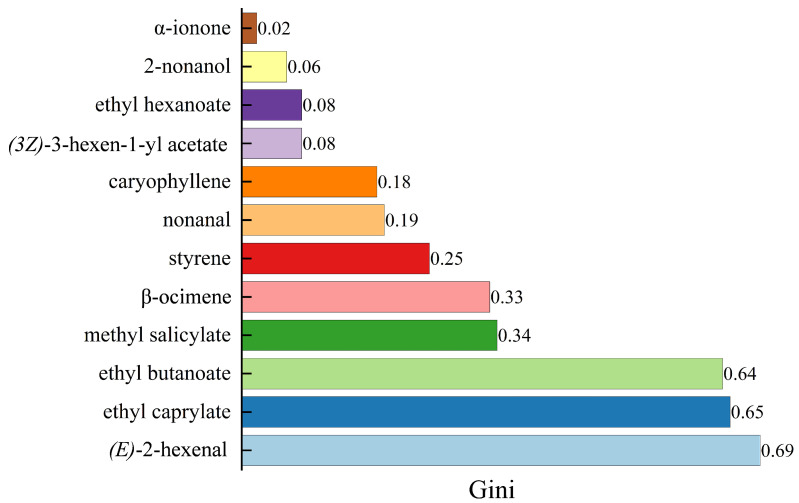
Gini graph of key flavor components.

**Figure 5 molecules-28-07879-f005:**
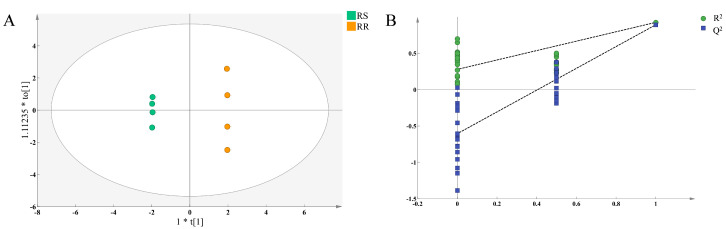
Comparison of the key flavor-contributing compounds in RR and RS. (**A**) OPLS-DA score plot: R2X = 0.92, R2Y = 1, Q2 = 0.978; (**B**) cross-validation plot for the OPLS-DA model with 200 calculations in a permutation test: R2 = (0.0, 0.279), Q2 = (0.0, –0.606).

**Table 1 molecules-28-07879-t001:** Relevant information and relative contents of VOCs.

No.	Compound	Formula	Retention Time (min)	CAS	Relative Content (%) ^A^	
					RS	RR
Alcohols (4 kinds)						
C1	ethanol	C_2_H_6_O	0.444	64-17-5	-	3.74 ± 1.42
C2	2-nonanol	C_9_H_20_O	9.817	628-99-9	0.91 ± 0.74	1.10 ± 0.74
C3	α-copaene	C_9_H_11_ClO	19.948	1000360-33-0	-	4.18 ± 3.81
C4	dihydro-β-ionol	C_13_H_24_O	22.150	3293-47-8	0.43 ± 0.51	-
Ether (1 kind)						
C5	(–)-dihydroedulan ii	C_13_H_22_O	17.029	41678-32-4	0.22 ± 0.18	-
Aldehydes (3 kinds)						
C6	(*E*)-2-hexenal	C_6_H_10_O	1.150	6728-26-3	47.88 ± 13.87	3.51 ± 0.58
C7	benzaldehyde	C_7_H_6_O	5.610	100-52-7	2.71 ± 1.58	-
C8	nonanal	C_9_H_18_O	10.611	124-19-6	0.81 ± 0.87	1.00 ± 0.63
Acids (3 kinds)						
C9	hexanoic acid	C_6_H_12_O_2_	6.591	142-62-1	11.71 ± 9.80	-
C10	butane-2,3-diyl diacetate	C_8_H_14_O_4_	8.14	1114-92-7	-	2.03 ± 1.09
C11	octanoic acid	C_8_H_16_O_2_	13.184	124-07-2	1.61 ± 1.34	1.47 ± 0.99
Esters (10 kinds)						
C12	ethyl butanoate	C_6_H_12_O_2_	0.297	105-54-4	-	3.97 ± 8.06
C13	ethyl acetate	C_4_H_8_O_2_	0.751	141-78-6	-	14.46 ± 5.7
C14	ethyl tiglate	C_7_H_12_O_2_	4.911	5837-78-5	-	0.84 ± 1.34
C15	ethyl hexanoate	C_8_H_16_O_2_	6.774	123-66-0	-	5.46 ± 2.89
C16	(*3Z*)-3-hexen-1-yl acetate	C_8_H_14_O_2_	7.425	3681-71-8	-	2.85 ± 1.38
C17	hex-2-enoic acid ethyl ester	C_8_H_14_O_2_	8.352	1552-67-6	-	0.18 ± 0.17
C18	sec-heptyl acetate	C_9_H_18_O_2_	8.547	5921-82-4	1.92 ± 0.96	-
C19	ethyl benzoate	C_9_H_10_O_2_	13.07	93-89-0	-	0.47 ± 0.15
C20	ethyl caprylate	C_10_H_20_O_2_	13.557	106-32-1	-	3.44 ± 2.21
C21	methyl salicylate	C_8_H_8_O_3_	13.855	119-36-8	0.46 ± 0.98	-
Alkanes (1 kind)						
C22	tetradecane	C_14_H_30_	20.732	629-59-4	0.14 ± 0.08	-
Terpenoids (28 kinds)						
C23	β-ocimene	C_10_H_16_	8.569	3338-55-4	0.94 ± 0.48	-
C24	theaspirane	C_13_H_22_O	17.306	36431-72-8	0.46 ± 1.33	-
C25	α-cubebene	C_15_H_24_	18.948	17699-14-8	0.09 ± 0.21	1.47 ± 2.69
C26	ylangene	C_15_H_24_	19.805	14912-44-8	-	0.11 ± 0.07
C27	α-ionol	C_13_H_22_O	20.019	25312-34-9	0.77 ± 0.69	-
C28	(–)-β-bourbonene	C_15_H_24_	20.131	5208-59-3	-	0.30 ± 0.91
C29	germacrene d	C_15_H_24_	20.430	23986-74-5	-	1.06 ± 1.25
C30	β-copaene	C_15_H_24_	20.445	18252-44-3	-	1.17 ± 0.33
C31	β-maaliene	C_15_H_24_	21.039	489-29-2	-	0.18 ± 0.12
C32	(–)-α-gurjunene	C_15_H_24_	21.051	489-40-7	-	0.56 ± 0.31
C33	caryophyllene	C_15_H_24_	21.234	87-44-5	3.56 ± 2.28	0.61 ± 1.62
C34	α-ionone	C_13_H_20_O	21.593	127-41-3	0.33 ± 0.26	-
C35	(+)-epi-bicyclosesquiphellandrene	C_15_H_24_	21.64	54274-73-6	-	0.42 ± 0.44
C36	valencene	C_15_H_24_	22.111	4630-07-3	1.67 ± 1.60	0.46 ± 2.55
C37	cubenene	C_15_H_24_	22.188	16728-99-7	-	0.30 ± 0.20
C38	cis-muurola-4(15),5-diene	C_15_H_24_	22.308	157477-72-0	-	0.38 ± 0.33
C39	γ-muurolene	C_15_H_24_	22.658	30021-74-0	-	0.38 ± 0.41
C40	epizonarene	C_15_H_24_	23.038	41702-63-0	-	1.45 ± 0.44
C41	δ-cadinene	C_15_H_24_	23.085	483-76-1	-	16.16 ± 8.15
C42	selina-4,11-dien	C_15_H_24_	23.328	103827-22-1	7.83 ± 6.99	7.21 ± 3.15
C43	β-selinene	C_15_H_24_	23.414	17066-67-0	1.78 ± 1.38	1.84 ± 0.50
C44	α-vetivenen	C_15_H_22_	23.529	28908-26-1	1.02 ± 0.83	-
C45	3,5,11-eudesmatriene	C_15_H_22_	23.543	193615-07-5	1.47 ± 0.70	0.85 ± 0.24
C46	α-muurolene	C_15_H_24_	23.843	31983-22-9	-	2.53 ± 1.84
C47	(*r*)-γ-cadinene	C_15_H_24_	24.262	39029-41-9	-	1.11 ± 0.74
C48	(–)-α-panasinsen	C_15_H_24_	24.364	56633-28-4	1.37 ± 1.85	2.01 ± 1.35
C49	cadinadiene	C_8_H_4_	24.80	29837-12-5	-	1.17 ± 0.39
C50	α-agarofuran	C_15_H_24_O	25.190	5956-12-7	0.14 ± 0.11	-
Aromatics (9 species)						
C51	styrene	C_8_H_8_	3.411	100-42-5	7.80 ± 1.91	3.75 ± 3.22
C52	4-methoxystyrene	C_9_H_10_O	12.353	637-69-4	-	0.31 ± 0.28
C53	estragole	C_10_H_12_O	14.198	140-67-0	-	1.77 ± 0.89
C54	anethole	C_10_H_12_O	17.112	104-46-1	-	0.30 ± 0.20
C55	α-calacorene	C_15_H_20_	25.118	21391-99-1	-	0.81 ± 0.71
C56	elemicin	C_12_H_16_O_3_	25.495	487-11-6	-	0.06 ± 0.04
C57	β-calacorene	C_15_H_20_	25.712	50277-34-4	-	0.09 ± 0.05
C58	α-corocalene	C_15_H_20_	27.422	20129-39-9	-	0.09 ± 0.06
C59	cadalin	C_15_H_18_	28.887	483-78-3	-	0.15 ± 0.15
Others (2 kinds)						
C60	cis-muurola-3,5-diene	C_17_H_22_N_4_O	22.185	1000365-95-4	-	0.26 ± 0.19
C61	*Z,Z,Z*-1,5,9,9-tetramethyl-1,4,7-cycloundecatriene	C_13_H_10_O	22.402	1000062-61-9	1.98 ± 2.31	2.01 ± 1.86

^A^ the relative content of VOCs is expressed as an average value ± standard deviation; “-” information was not found in the literature; relative content: refer to Section 4.5 for calculations, indicated by “mean ± standard deviation (SD)”; RS: *Rosa roxburghii* Tratt.; RR: *Rosa sterilis*.

**Table 2 molecules-28-07879-t002:** The ROAVs of key flavor components.

No.	Compound	*T* (mg/kg)	Odor Description	ROAVs	
				RS	RR
1	2-nonanol	0.07	Waxy, creamy, citrus, orange, cheese, fruity	1.76	0.36
2	(*E*)-2-hexenal	0.4286	green banana, fatty, cheesy	15.15	0.19
3	nonanal	0.0011	Waxy, rose, fresh orris, orange peel, fatty	100.00	20.64
4	ethyl butanoate	0.0009	fruity, pineapple, brandy	<0.1	100.00
5	ethyl hexanoate	0.005	sweet, fruity, pineapple, waxy, green banana	<0.1	24.76
6	(*3Z*)-3-hexen-1-yl acetate	0.031	sweet, fruity, green banana, apple, grassy	<0.1	2.08
7	ethyl caprylate	0.0193	Fruity, wine, waxy, sweet, apricot, green banana, brandy, pear	<0.1	4.04
8	methyl salicylate	0.04	Wintergreen, mint	1.57	<0.1
9	β-ocimene	0.034	Floral, herb, flower, sweet	3.75	<0.1
10	caryophyllene	0.064	sweet, woody, spice, clove	7.54	0.22
11	α-ionone	0.0106	Sweet, woody, floral, violet orris, fruity	4.19	<0.1
12	styrene	0.065	sweet, balsam, floral, plastic	16.26	1.31

Odor descriptions were cited from http://www.thegoodscentscompany.com; “*T*” was taken from a book titled “*Compilations of odor threshold values in air, water and other media*” [30]; ROAVs: relative odor activity values; RS: *Rosa roxburghii* Tratt.; RR: *Rosa sterilis*.

**Table 3 molecules-28-07879-t003:** VIP and *X_A_* of key flavor components.

No.	Compound	OPLS-DA	RF	*X_A_*
		VIP	Gini	
C1	(*E*)-2-hexenal	1.52633	0.69	1.00
C2	ethyl caprylate	1.50715	0.65	0.96
C3	β-ocimene	1.22437	0.33	0.62
C4	ethyl butanoate	1.12642	0.64	0.82
C5	styrene	0.945676	0.25	0.46
C6	(*3Z*)-3-hexen-1-yl acetate	0.92031	0.08	0.33
C7	methyl salicylate	0.908079	0.34	0.52
C8	ethyl hexanoate	0.905574	0.08	0.32
C9	nonanal	0.760134	0.19	0.35
C10	α-ionone	0.628868	0.02	0.18
C11	caryophyllene	0.502894	0.18	0.25
C12	2-nonanol	0.137078	0.06	0.03

OPLS-DA: orthogonal partial least squares discriminant analysis; VIP: importance in the projection; RF: random forest; Gini is the result of RF computation; *X_A_*: refer to Section 4.6 for calculations.

## Data Availability

The data supporting the results of this study are included in the present article.

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
