# Peer review of "Analysis of Volatile Components in Rosa roxburghii Tratt. and Rosa sterilis Using Headspace–Solid-Phase Microextraction–Gas Chromatography–Mass Spectrometry"

_molecules, 2023, doi:10.3390/molecules28237879_

Round 1

Reviewer 1 Report (Previous Reviewer 1)

Comments and Suggestions for Authors

Dear authors, my opinion is that your results are very similar to published before. So I think that results are not novel

Author Response

Thanks for the reviewer's comment. We will search the references more thorough and emphasize the novelty of the study in the future. We hope to capture your interest with our future articles.

Reviewer 2 Report (Previous Reviewer 2)

Comments and Suggestions for Authors

Dear Editor

Authors have significantly improved the manuscript entitled "Analysis of Volatile Components in Rosa roxburghii Tratt. and Rosa sterilis Using HS-SPME-GC-MS". This study showed interesting information for the researchers and scientists. I have thoroughly checked the revised version of the manuscript. In the present form it can be accepted for publication in the journal.

Authors suggested to revise the key words, these should be different form title

Author Response

Thanks for the reviewer's comment and recognition of this work. Your valuable suggestions during the review process helped we better the article. Hope everything goes well with your work. Meanwhile, we have supplemented and improved the Keywords part in the revised manuscript (Line 28).

Reviewer 3 Report (Previous Reviewer 3)

Comments and Suggestions for Authors

Thanks for your response. I think this work could be accepted in the current version.

Author Response

Thanks for the reviewer's comment and recognition of this work. Your valuable suggestions during the review process helped we better the article. Hope everything goes well with you.

Reviewer 4 Report (Previous Reviewer 4)

Comments and Suggestions for Authors

The remarks raised were taken into consideration by the authors therefore it can be accepted

Author Response

Thanks for the reviewer's comment and recognition of this work. Your valuable suggestions during the review process helped we better the article. Hope everything goes well with you.

Reviewer 5 Report (Previous Reviewer 5)

Comments and Suggestions for Authors

The authors have responded to all my comments and suggestions. As such, the manuscript can be acceptable for publication in its current status.

Author Response

Thanks for the reviewer's comment and recognition of this work. Your valuable suggestions during the review process helped we better the article. Hope everything goes well with you.

This manuscript is a resubmission of an earlier submission. The following is a list of the peer review reports and author responses from that submission.

Round 1

Reviewer 1 Report

Comments and Suggestions for Authors

Dear authors,

 I found many articles which are not cited with very similar  investigations and in fact are the part of this investigation, for example

Niu Y, Wang R, Xiao Z, Sun X, Wang P, Zhu J, Cao X. Characterization of Volatile Compounds of Rosa roxburghii Tratt by Gas Chromatography-Olfactometry, Quantitative Measurements, Odor Activity Value, and Aroma Intensity. Molecules. 2021 Oct 14;26(20):6202. doi: 10.3390/molecules26206202. PMID: 34684797; PMCID: PMC8539914.

Sheng, Xiaofang & Huang, Mingzheng & Li, Tingting & Li, Xin & Cen, Shunyou & Li, Qinyang & Huang, Qun & Tang, Weiyuan. (2023). Characterization of aroma compounds in Rosa roxburghii Tratt using solvent-assisted flavor evaporation dspace-solid phase microextraction coupled with gas chromatography-mass spectrometry and gas chromatography-olfactometry. Food Chemistry: X. 18. 100632. 10.1016/j.fochx.2023.100632. 

Liu, M.-H.; Zhang, Q.; Zhang, Y.-H.; Lu, X.-Y.; Fu, W.-M.; He, J.-Y. Chemical Analysis of Dietary Constituents in Rosa roxburghii and Rosa sterilis Fruits. Molecules 201621, 1204. https://doi.org/10.3390/molecules21091204

So, in my opinion, novelty of this investigation is insufficient. 

As presented I think it should be rejected

Reviewer 2 Report

Comments and Suggestions for Authors

Dear Editor

Its very interesting study to Analysis of Volatile Components in Rosa roxburghii Tratt. and Rosa sterilis Using HS-SPME-GC-MS techniques. Its is very normal technique to identify the volatiles in different tissue. There is need to improve to manuscript quality before final acceptance.

Introduction part is very weak, too short, author should add more details about previous work on rose flowers. Add more information, importance or relevance of this study. there is plenty of literature available for references (

Comparative analysis of flower volatiles from nine citrus at three blooming stages

Citrus leaf volatiles as affected by developmental stage and genetic type.

The Sensory Quality Improvement of Citrus Wine through Co-Fermentations with Selected Non-Saccharomyces Yeast Strains and Saccharomyces cerevisiae.

Floral Scent Chemistry of Luculia yunnanensis (Rubiaceae), a Species Endemic to China with Sweetly Fragrant Flowers.

Author need to highlight the importance of this study at the end of introduction part.

Figure 2 is not clear, redesign again with clear information.

Table 3, abbrrevations and other details need to be mention

Discussion part is too short and missing a lot information in it. Author suggested to revise this part carefully and establish the link of their findings. there should be some clear information about genetic or other role for their investigation of this study or major identified compounds and their role in the environment. Conclusion should be revise rigorously as it lack main idea of the manuscript. Author should write material and method in details.

I strong recommend Major Revision of this paper to improve the quality for this journal

Comments on the Quality of English Language

Dear Editor

Its very interesting study to Analysis of Volatile Components in Rosa roxburghii Tratt. and Rosa sterilis Using HS-SPME-GC-MS techniques. Its is very normal technique to identify the volatiles in different tissue. There is need to improve to manuscript quality before final acceptance.

Introduction part is very weak, too short, author should add more details about previous work on rose flowers. Add more information, importance or relevance of this study. there is plenty of literature available for references (

Comparative analysis of flower volatiles from nine citrus at three blooming stages

Citrus leaf volatiles as affected by developmental stage and genetic type.

The Sensory Quality Improvement of Citrus Wine through Co-Fermentations with Selected Non-Saccharomyces Yeast Strains and Saccharomyces cerevisiae.

Floral Scent Chemistry of Luculia yunnanensis (Rubiaceae), a Species Endemic to China with Sweetly Fragrant Flowers.

Author need to highlight the importance of this study at the end of introduction part.

Figure 2 is not clear, redesign again with clear information.

Table 3, abbrrevations and other details need to be mention

Discussion part is too short and missing a lot information in it. Author suggested to revise this part carefully and establish the link of their findings. there should be some clear information about genetic or other role for their investigation of this study or major identified compounds and their role in the environment. Conclusion should be revise rigorously as it lack main idea of the manuscript. Author should write material and method in details.

I strong recommend Major Revision of this paper to improve the quality for this journal

Reviewer 3 Report

Comments and Suggestions for Authors

This work investigated volatile organic compounds (VOCs) and flavor characteristics of Rosa roxburghii Tratt. (RR) and Rosa sterilis (RS) using headspace solid-phase microextraction coupled with gas chromatography-mass spectrometry (HS-SPME-GC-MS). Differences VOCs between RR and RS could provide a basis for further development and utilization.

The two RR and RS were investigated. What the major difference of the two fruits in the application? The authors should illustrate why the two fruits were taken into consideration. From the results, VOCs are different in the two fruits. The authors should also discuss the major differences in RR and RS. Which one could be used for the evaluation of fruit quality in the following utilization. Section 4.5 Please explain how to calculate the values of T, OAVi, OAVmax, and C. Is anyone a constant value? Section 2.1. Figure 1 could be moved to supplementary materials since there are limit information in the figure. Table 1. The authors should explain how to calculate the relative content in the table.

Reviewer 4 Report

Comments and Suggestions for Authors

Comments on the Quality of English Language

The manuscript is well written and the English is good

Reviewer 5 Report

Comments and Suggestions for Authors

The current manuscript can be accepted for publication on condition that the authors respond to the following comments and inquiries. Upon receiving the authors' response, the manuscript can be accepted for publication.

1.      Provide the pharmacological importance of the volatile organic compounds (major) observed in the study concerning in-vivo and in-vitro models.

2.      The introduction needs to be elaborated on the previously reported studies on Rosa spp.

3.      Authors should include relative references corresponding to GC-Head space analysis. For example., Ref Deeper Insight into the Volatile Profile of Rosa willmottiae with Headspace Solid-Phase Microextraction and GC–MS Analysis. 10.3390/molecules2704124010.3390/molecules27041240.

4.      Although the results were summarized properly, how would the authors conclude the study's novelty from the previously reported similar studies using GC-MS or HS-GC-MS analysis?

5.      Discuss the substantial method development parameters that would impact the study's significance.

6.      Provide the method development protocols tested for the study in the optimization process using SPE-HS analysis as Supporting information.

7.      Can the volatile compound profiles indicate the quality or authenticity of these roses products in the market?

8.      Are there any novel volatile compounds identified in these rose species that have not been reported in other related plants?